# Inter-Session Reliability of Functional Near-Infrared Spectroscopy at the Prefrontal Cortex While Walking in Multiple Sclerosis

**DOI:** 10.3390/brainsci10090643

**Published:** 2020-09-17

**Authors:** Kim-Charline Broscheid, Dennis Hamacher, Juliane Lamprecht, Michael Sailer, Lutz Schega

**Affiliations:** 1Department of Sport Science, Institute III, Otto von Guericke University (OvGU) Magdeburg, Zschokkestraße 32, 39104 Magdeburg, Germany; Dennis.Hamacher@dhgs-hochschule.de (D.H.); lutz.schega@ovgu.de (L.S.); 2MEDIAN Neurological Rehabilitation Center Magdeburg, Gustav-Ricker-Straße 4, 39120 Magdeburg, Germany; Juliane.Lamprecht@median-kliniken.de (J.L.); Michael.Sailer@median-kliniken.de (M.S.); 3Institute for Neurorehabilitation affiliated to OvGU Magdeburg, MEDIAN Neurological Rehabilitation Center Magdeburg, Gustav-Ricker-Straße 4, 39120 Magdeburg, Germany; 4MEDIAN Rehabilitation Center Flechtingen, Parkstraße, 39345 Flechtingen, Germany

**Keywords:** MS, hemodynamic response, fNIRS, test–retest reliability, cortical activity, PFC

## Abstract

Many established technologies are limited in analyzing the executive functions in motion, especially while walking. Functional near-infrared spectroscopy (fNIRS) fills this gap. The aim of the study is to investigate the inter-session reliability (ISR) of fNIRS-derived parameters at the prefrontal cortex while walking in people with multiple sclerosis (MS) and healthy control (HC) individuals. Twenty people with MS/HC individuals walked a 12 m track back and forth over 6 min. The primary outcomes were the absolute and relative reliability of the mean, slope coefficient (SC), and area under the curve (A) of the oxy-/deoxyhemoglobin concentrations (HbO/HbR) in the Brodmann areas (BA) 9/46/10. The SC and the A of HbO exhibited a fair ISR in BA10 in people with MS. For the mean and A of the HbR, almost all areas observed revealed a fair ISR. Overall, the ISR was better for HbR than HbO. A fair to excellent ISR was found for most BA of the prefrontal cortex in HC individuals. In total, the ISR of the analyzed fNIRS-derived parameters was limited. To improve the ISR, confounders such as fatigue and mind wandering should be minimized. When reporting the ISR, the focus should be on the mean/A rather than SC.

## 1. Introduction

Human bipedal locomotion is a central determinant of participation in daily life. Especially people suffering from inflammatory autoimmune diseases, such as multiple sclerosis (MS), often exhibit impaired locomotion [1]. These impairments can be diverse (e.g., ataxia, spasticity, or muscle weakness) and depend on the affected area in the brain or spinal cord [2]. To treat these deficits more efficiently, it is necessary to understand the underlying motor and cognitive mechanisms. 

One concept that comprises both mechanisms is gait automaticity. According to Clark [3], gait automaticity is “[…] the ability of the nervous system to successfully coordinate movement with minimal use of attention-demanding executive control resources”. The interaction of automaticity and executive control are essential for executing movements. The respective contributions and the relation to each other can be shifted by different factors such as (motor-) learning progresses [4] or damage of the central nervous system [3], as it can be found in people with MS. To estimate the degree of gait automaticity, it is important to quantify the activation of the prefrontal cortex (PFC) in which the executive functions and the attention are located [5]. 

The established technologies (magnetic resonance imaging, positron emission tomography, and magnetoencephalography) are too limited to assess the PFC activation in motion due to the required fixed head position and non-portability [6]. Although electroencephalography (EEG) is portable, the preparation is time-consuming, and it has a high susceptibility to motion artifacts [6]. 

Functional near-infrared spectroscopy (fNIRS) is a promising tool that provides the following properties [7]. It is a non-invasive, easy to apply, and portable optical brain imaging method that is applicable in motion [8,9,10]. It is less affected by motion artefacts than comparable systems and has a relatively high temporal resolution up to 1 ms [8,11]. Due to these features, the interest in fNIRS is rapidly increasing in the rehabilitative context. First studies have already investigated the PFC activation while walking in people with MS [12,13]. They were able to distinguish between healthy people and people with MS and between different walking conditions based on the PFC activation. Even though these first results are promising, there is a lack of basic methodological studies on fNIRS.

To the best of our knowledge, there are no studies analyzing absolute and relative inter-session reliability (ISR) data in people with MS while walking yet, which is urgently necessary to assess changes in fNIRS-derived parameters. Especially in people with MS, it is important to verify the ISR, as the disease is accompanied by a high fluctuation in daily performance caused by, e.g., state fatigue or fatigability [14]. Moreover, there is only one study that has demonstrated moderate ISR of fNIRS-derived parameters while walking in healthy adults [15] yet. 

Therefore, the present study aims to analyze the relative and absolute ISR of fNIRS-derived parameters at the PFC during single-task walking on two consecutively days in moderately affected people with MS and healthy control individuals (HC individuals).

## 2. Materials and Methods

### 2.1. Study Design and Participants

For this cross-sectional controlled ISR study, 20 people with MS (15 female/5 male) with a confirmed MS diagnosis according to the revised McDonald criteria [16] were recruited. They were 41.0 ± 12.0 years old and had an Expanded Disability Status Scale (EDSS) [17] of 2.0 ± 0.9. The patients had to be able to walk at least 300 m without walking aids. Therefore, only patients with an EDSS less than or equal to 4.5 were included. The last acute episode of MS and the last cortisone intake should date back more than 30 days. The HC individuals were age- (42.2 ± 9.8 years) and sex- (16 female/4 male) matched. They should not have orthopedic or neurologic limitations nor hypertension or obesity. The study was approved by the ethics committee of the Medical Faculty of the Otto von Guericke University (OvGU) Magdeburg (Germany) (No.: 116/18) and is registered in the German Clinical Trial Register (ID: DRKS00015190).

### 2.2. Study Procedure

The study was conducted by the Department of Health and Physical Activity of the OvGU Magdeburg together with the Center for Neurorehabilitation Median Klinik Flechtingen (Germany). The people with MS were recruited by health professionals at the clinic at the beginning of their six weeks rehabilitation. First, the patients were informed about the study and written informed consent was obtained. In the pre-assessment, the 6-min walk test (6MWT) was executed [18] by physiotherapists and the 12-Item Multiple Sclerosis Walking Scale (MSWS-12, German version) [19] was obtained. Subsequently, the test and retest measurements (24 h in between) were conducted in the morning on non-treatment days. The participants walked a distance of 12 m on a level floor back and forth, in their self-selected walking pace and were advised to concentrate on walking only. Due to the fNIRS system requirements, the measurement started in a standing position (baseline) for 30 s and then altered between standing and walking every 30 s (Figure 1). The test conditions standing and walking were announced by the test instructor. The number of walking intervals was chosen according the time of the 6MWT. In total, the duration of the test protocol was about 12 min and 30 s. At the beginning and the end of each test day, the subjects were asked about their perceived exhaustion using the Borg Scale [20].

The HC individuals were recruited from local citizens. The measurements were conducted at the facilities of the OvGU Magdeburg. The test procedure was the same as for the people with MS.

### 2.3. Equipment and Outcome Measures

For this study, two portable fNIRS systems (NIRSport, NIRx Medical Technologies, NY, USA) were used each attached to a standardized cap (EasyCap GmBH, Herrsching, Germany) with circumferences of 56 cm and 58 cm. Each cap was equipped with eight sources and eight detectors together with eight short separation channels according to the international 10–20 system for EEG to cover the PFC (Prefrontal cortex) (Figure 2: created with NirSite 2.0, NIRx Medical Technologies, NY, USA). The average source-detector separation distance was 30–40 mm. The arrangement of the optodes was done with the fNIRS Optodes’ Location Decider (fOLD) toolbox [21]. Additional information about the sensitivity of the channels according to the fOLD toolbox is provided in the Appendix A.

The cap was placed in the middle between nasion to inion and left preauricular to right preauricular point (reference point Cz). To deal with external light interferences an additional standardized cap was placed on top of the fNIRS system. The applied fNIRS system operates at two different wavelengths (760/850 nm) and at a fixed sampling frequency of 7.81 Hz. The exact subareas captured are the right, left, and medial dorsolateral PFC Brodmann area (BA) 9 and 46 (r/lDLPFC9, r/lDLPFC46, mDLPFC9) and the right, left, and medial frontopolar cortex BA10 (r/l/mFPC10). The subareas are composed of the following channels: rDLPFC9 (channels, 1, 18 and 21), rDLPFC46 (channel 6), lDLPFC9 (channels 17, 20 and 22), lDLPFC46 (channel 13), rFPC10 (channels 4, 5, 7 and 8), lFPC10 (10, 11, 12 and 14), and mFPC (channel 9).

The primary outcomes were the concentration of oxy-/deoxyhemoglobin (cHbO/cHbR) in those subareas. The secondary outcomes were the heart rate (HR) and heart rate variability (HRV) measured with a heart rate monitor (RS800CX Polar Electro Oy ^®^, Kempele, Finland). The HRV parameters considered were the time intervals between two R-spikes (RR interval) and the low frequency/high frequency (LF/HF) ratio. HR and HRV were used to control systemic confounders in the hemodynamic response [22]. Additionally, the perceived exhaustion was assessed on both days pre and post measurement using the Borg Scale (rating 6–20).

### 2.4. Data Processing

For data processing, we used the software “HOMER2” Version 2.8 [23]. First, the data were processed with the enPruneChannels function to sort out the channels with a too weak or too strong signal or where the standard deviation was too high (data range: 1 × 10^−2^ to 1 × 10^7^; signal to noise threshold: 2; source detector separation range: 0.0–45.0 mm, and reset: 0). Subsequently, the raw data were transformed to optical density data [23]. The second filter method was utilized to reduce motion artefacts based on a spline interpolation and the digital Savitzky-Golay filter (hmrMotionCorrectSplineSG) [24]. Therefore, the *p* value was set to 0.99 [24]. The frame size was adjusted to 15 s. The data were then processed with a 3rd order Butterworth low pass filter with a cut off frequency of 0.5 Hz [24,25]. Consecutively, the filtered optical density data were converted into the changes in cHbO/cHbR by executing the modified Beer–Lambert Law [10]. To incorporate the age-related differences, the differential path length factor was adjusted, as described in [26], for each participant. The hemodynamic response function (HRF) was appraised by a general linear model approach. Therefore, the ordinary least squares method was used [27]. The time range was set from −10 to 45 s. The basis function for the HRF is a consecutive sequence of Gaussian functions with the width of 0.5 and the temporal spacing of 0.5. For the baseline drift, a 3rd order polynomial drift correction was utilized. The regression was conducted with the nearest short separation channels. After these preprocessing steps, the block average was calculated. 

The cHbO/cHbR obtained during the walking protocol (twelve times 30 s) was further processed in MATLAB (Version R2017b, The MathWorks, Natick, MA, USA). To illustrate the course of cHbO and cHbR from baseline through walking to the next baseline, the channels of each individual subject were first averaged to the corresponding subareas of the PFC (l/r/mDLPFC9/46 and l/r/mFPC10). Then, the mean and standard deviation were calculated over all subjects for the respective subareas. Here, the last 10 s of the previous baseline, the 30 s walking interval and 15 s of the succeeding baseline were included to get an impression of the signal’s increase and decrease. 

To prepare the data for the absolute and relative ISR calculation, the cHbO and cHbR were averaged from all twelve walking intervals of 30 s each. The first and last 5 s were cut out due to the delay of the hemodynamic response at the beginning and to reduce possible influences of the expected end of the walking interval. Subsequently, the mean, the slope coefficient (SC) [28], and the area under the curve (A) [29] of the cHbO and cHbR of this interval (5–25 s) were calculated. The mean and the A have been applied frequently in literature [28]. The SC provides information about the magnitude and direction of the change in cHbO and cHbR and is determined by a linear regression method [28]. 

### 2.5. Statistical Analysis

The statistical analysis was performed with the IBM SPSS software (Statistical Package for social science, Version 25, Chicago, IL, USA). The normal distribution was verified using the Kolmogorov–Smirnov test. The relative ISR was determined by the intraclass correlation coefficient (ICC) estimates and their 95% confidence intervals (CI) of the mean, SC, and A of the cHbO/cHbR build on a single-rating, absolute-agreement, 2-way, mixed-effects model [30]. The ICC was classified as poor with values ≤ 0.40, fair between 0.40 and 0.59, good between 0.60 and 0.74, and excellent between 0.75 and 1.00 [31]. In addition, the absolute reliability was checked by applying Bland and Altman limits of agreement (LoA), the bias, and the CI of the lower and upper LoA [32]. The differences of the secondary outcomes between testing days were tested by paired t-tests or, in case of none, normal distribution by Wilcoxon tests.

## 3. Results

The data of 16 people with MS (14 female/2 male) and 19 HC individuals (15 female/4 male) with an average age of 41.0 ± 12.0 and 42.1 ± 9.8 years, respectively, were analyzed (Table 1). Four people with MS had to be excluded due to an acute episode during the study period, breathing problems (allergic coryza) during the measurement, and two for not finishing the measurement. One subject of the HC individuals had to be excluded due to obesity (body mass index: 36.5). Overall, the people with MS suffered from moderate walking limitations (MSWS-12: 45% ± 20.7%) and were able to cover 473.1 ± 109.7 m in the 6MWT (HC individuals: 533.5 ± 64.5 m).

### 3.1. Descriptive Data cHbO/cHbR

We found the highest cHbO in the l/rDLPFC46 on both days in people with MS (Table 2). The cHbR in the lDLPFC46 was lowest ranging from −0.047 to −0.036 µmol/L. In the rDLPFC46 the cHbR varied greatly between testing days in people with MS (test: −0.014 ± 0.057 µmol/L; retest: −0.081 ± 0.122 µmol/L). The only negative cHbO was found in the mFPC10 for all people with MS on both testing days ranging from −0.103 to −0.024 µmol/L.

We further observed that the cHbO were in general close to zero and mostly negative in HC individuals. The only positive results and with it the highest activation while walking were recorded for the l/rDLPFC46. 

As illustrated exemplarily in Figure 3, the mean and standard deviation of the cHbO in the lDLPFC46 is higher on test than on retest day across all subjects (HC individuals: *n* = 19/MS: *n* = 16). 

In addition, the overall mean cHbO (Table 2) indicated also a trend that in some subareas the activation was lower on the second compared to the first day for both groups. Especially the mean cHbO in the lDLPFC9/46 and rDLPFC46 in people with MS and in the l/rDLPFC9, lDLPFC46, and l/rFPC10 in HC individuals revealed this trend. For further details regarding the cHbO and cHbR, please see Table 2.

### 3.2. Inter-Session Reliability cHbO/cHbR

All results regarding the ISR are listed in Table 3. For the people with MS, no ISR for the mean of the cHbO could be proven. The SC of the cHbO for the l/mFPC10 (ICC = 0.54/0.58) and the A of the cHbO for the rFPC10 (ICC = 0.42) exhibited a fair ISR. Regarding the cHbR, a fair ISR for all subareas (ICC range = 0.46-0.56) except the r/lDLPFC9 (ICC = 0.39/0.36) could be determined. The ISR of the A of the cHbR was comparable. For the SC of the cHbR, a fair ISR for the l/mFPC10 (ICC = 0.47/0.40) and the lDLPFC46 (ICC = 0.40) and a good ISR for the rFPC10 (ICC = 0.63) could be demonstrated. 

In the HC individuals, almost all subareas displayed at least a fair ISR of the mean and A of the cHbO except the mDLPFC9 (mean/A: ICC = 0.39) and the rFPC10 (A: ICC = 0.37). Furthermore, the l/mFPC10 and lDLPFC9 showed a good ISR for the mean cHbO and the lDLPFC9 an excellent ISR for the A of the cHbO. The SC of the cHbO exhibit a fair ISR for l/rDLPFC9 and r/mFPC10. The mean and the A of the cHbR were comparable concerning the ICC. In both cases, almost all subareas demonstrated a fair ISR except the mFPC10 (ICC = 0.39) with a poor and the lFPC10 with a good (ICC = 0.63) ISR. Concerning the A of the cHbR, the rFPC10 displayed also a good ISR (ICC = 0.62). The ICC for the A and the mean of the cHbO in the rDLPFC9 and for the A of the cHbR in the rFPC10 have to be interpreted with caution due to the non-normal distribution.

The highest bias/mean difference (Bland and Altman, Table 4) was shown in the lDLPFC46 (people with MS: 0.099 µmol/L; HC individuals: 0.046 µmol/L). The lowest bias was found in the rDLPFC46 (people with MS: 0.003 µmol/L) and mDLPFC9 (HC individuals: 0.000 µmol/L). For the mean cHbR the highest bias was identified in the rDLPFC46 (people with MS: 0.067 µmol/L) and mDLPFC9 (HC individuals: −0.016 µmol/L). The lowest was observed in the mDLPFC9 (people with MS: −0.001 µmol/L) and lDLPFC46 (HC individuals: 0.001 µmol/L), respectively. The LoA of the mean cHbO were smallest in mDLPFC9 (0.322/−0.361 µmol/L) in people with MS and lDLPFC9 (0.356/−0.280 µmol/L) as well as mFPC10 (0.311/−0.331 µmol/L) in HC individuals. Considering the LoA of the mean cHbR, these were narrowest in rDLPFC9 (people with MS: 0.092/−0.088 µmol/L) and lFPC10 (HC individuals: 0.084/−0.094 µmol/L). 

### 3.3. Secondary Outcomes

The mean HR did not differ significantly between test and retest in people with MS (test: 98.1 ± 12.2 bpm/retest: 97.3 ± 16.4 bpm) and in HC individuals (test: 91.1 ± 10.2 bpm/retest: 91.5 ± 8.2 bpm). In addition, the mean RR interval did not reveal any difference in people with MS (test: 618.09 ± 72.52 ms/retest: 626.37 ± 104.60 ms) and in HC individuals (test: 674.58 ± 80.24 ms/retest: 660:39 ± 59.63 ms) as well as the standard deviation of the RR interval (people with MS test: 19.51 ± 9.08 ms/retest: 22.75 ± 14.02 ms and HC individuals (test: 60.78 (44.20/90.78) ms/retest: 70.25 (45.77/128.56) ms)). Furthermore, no difference could be found for the mean LF/HF ratio in people with MS (test: 3.51 ± 2.06/retest: 4.67 ± 4.25) and HC individuals (test: 2.72 ± 1.06/retest: 2.77 ± 1.32).

The perceived exhaustion (Borg Scale) was rated as “very light” to “light” in people with MS (median (25%/75% quartile) test: 8 (8/10); retest: 8 (8/12)) and in HC individuals (test: 9 (8/11); retest: 10 (8/11)).

## 4. Discussion

Clinicians need precise diagnostic tools with reasonable reliability to be able to deduct specially tailored intervention strategies. Therefore, the aim of the study was to verify the ISR of the fNIRS-derived parameters HbO and HbR while single-task walking in people with MS and HC individuals.

Basically, a fair to excellent ISR of the fNIRS-derived parameters in the subareas of the PFC could be proven for the HC individuals in our study. Our results are partly congruent with those of Stuart et al. [15] who found a moderate ISR in the overall PFC.

In people with MS, the ISR was very limited in our study. One explanation could be that the daily performance of people with MS can fluctuate greatly mainly due to fatigue symptoms [14]. It is also known that the motor and cognitive performance of people with MS decline over the course of the day [14]. However, we have tried to keep these influences as low as possible by performing the measurement in the morning in a rested state and without prior treatment. We also checked the exhaustion state pre and post measurement on both days, and there was no difference between days nor before and after the walking trial in both groups.

An interesting outcome for both groups is a relatively high activation of the lDLPFC46 in comparison to the other subareas and the poor ISR. It is known that in the case of mind wandering during simple tasks the lDLPFC46 is involved [33]. Single-task walking might provoke mind wandering due to its low requirements. Mind wandering is not a constant factor between days and could be an explanation for the partly poor ISR in both groups.

Another interesting result is that the ICCs were higher for the cHbR than for the cHbO in people with MS. This is in line with the results of a study by Plichta et al. [34] that quantified ISR of fNIRS measures during different finger-tapping tasks. In contrast, other studies verifying the ISR of fNIRS for motor [35] or cognitive [36] tasks demonstrated that cHbO is more reliable than cHbR. However, these studies are only comparable to a limited extent, as different brain areas and cohorts were investigated. Nevertheless, an explanation for a better ISR of the HbR could be that it is less affected by physiological noise [35,37,38] and that it is spatially more concentrated [35,36] than HbO. Another explanation could be that HbO is more sensitive to changes in blood flow [37] and therefore may be more susceptible to fluctuations from day to day.

Overall, it has been shown that the ISR for the mean and A of cHBO and cHbR are comparable among each other. However, the ISR of the SC of the cHbO and cHbR was worse. The mean is known to be relatively robust against motion artefacts [39]. The SC was only reported in the context of cognitive tasks [28], and it is not yet clear how robust this parameter would be in regard to movement artefacts.

One limitation in the experimental procedure, which is perhaps responsible for the relatively high bias (Bland and Altman), is that the fNIRS cap was manually aligned using anatomical landmarks without any other technical aids. Nevertheless, the cap was always fitted by the same experienced investigator according to current 10–20 EEG system standards.

Furthermore, we have assumed that single-task walking does not need to be familiarized. However, in both groups, it was observed that the activation of some subareas of the PFC were lower on the retest than on the test day. Therefore, there might have been a certain learning effect even though the task was simple. A familiarization could have improved the reliability, as Hamacher et al. have already demonstrated for kinematic gait parameters [40].

In conclusion, it would be helpful for future studies (i) to control state fatigue in people with MS more adequately by applying, e.g., the Profile of Mood States questionnaire [41], (ii) to add an easy cognitive task guiding the attention to minimize possible mind wandering, (iii) to report the mean/A of the cHbO/cHbR rather than the SC, (iv) to improve the placement of the fNIRS cap by applying a 3D digitizer, and (v) to familiarize even very simple tasks.

## Figures and Tables

**Figure 1 brainsci-10-00643-f001:**
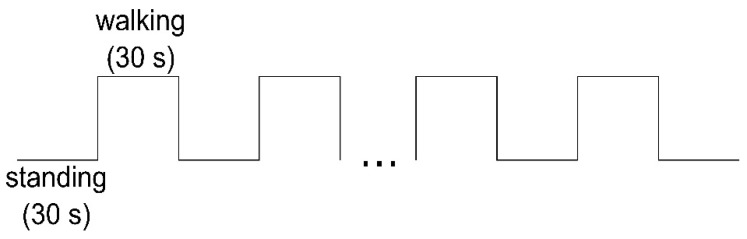
fNIRS test protocol.

**Figure 2 brainsci-10-00643-f002:**
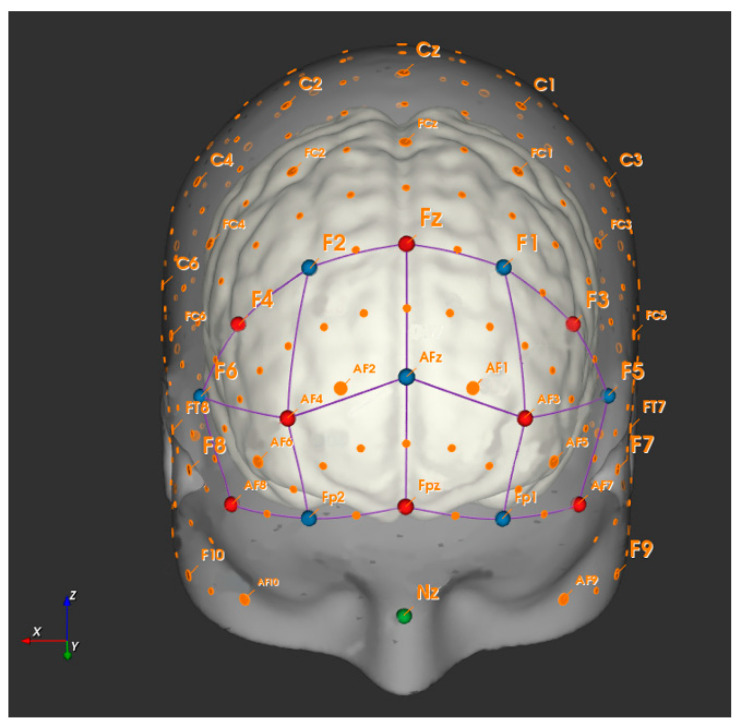
Arrangement of sources (red dots) and detectors (blue dots) at the prefrontal cortex using fNIRS.

**Figure 3 brainsci-10-00643-f003:**
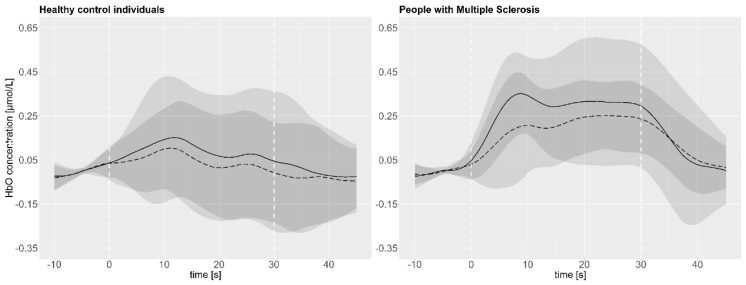
Mean cHbO test (continuous line) and retest (dashed line) data and their respective standard deviation (test: light grey/retest: dark grey) of the lDLPFC46 over all subjects over the entire course of the measurement; 30 s walking interval is indicated by the white dashed lines.

**Table 1 brainsci-10-00643-t001:** Descriptive subject data.

	Age [Years]	f/m	Weight [kg]	Height [cm]	EDSS	FD [Years]	FM [years]	MSWS-12 [%]	6MWT [m]
MS (*n* = 16)	41.0 ± 12.0	14/2	74.6 ± 18.1	170.1 ± 9.0	2.0 ± 0.9	5.9 ± 6.8	8.6 ± 8.7	45.0 ± 20.7	473.1 ± 109.7
HC (*n* = 19)	42.2 ± 9.8	15/4	73.0 ± 16.0	171.4 ± 8.8	n.a.	n.a.	n.a.	n.a.	533.5 ± 64.5

MS: multiple sclerosis; HC: healthy control; f: female; m: male; EDSS: Expanded Disability Status Scale; FD: first diagnosis; FM: first manifestation; MSWS-12: 12-Item Multiple Sclerosis Walking Scale; 6MWT: 6-min walk test; n.a.: not applicable.

**Table 2 brainsci-10-00643-t002:** Test and retest overall mean cHbO/cHbR for both groups (MS/HC).

		MS (*n* = 16)					HC (*n* = 19)				
	Area	Mean	MD	SD	25% Quartile	75% Quartile	Mean	MD	SD	25% Quartile	75% Quartile
Test HbO [µmol/L]	lDLPFC9	0.114 ^#^	0.090	0.214	−0.001	0.135	−0.083	−0.081	0.217	−0.204	0.056
	rDLPFC9	0.055 ^#^	−0.003	0.177	−0.057	0.154	−0.106 ^#^	−0.075	0.222	−0.283	0.025
	lDLPFC46	0.315 ^#^	0.304	0.207	0.212	0.351	0.104	0.074	0.257	−0.086	0.332
	rDLPFC46	0.210	0.201	0.132	0.111	0.290	0.067	0.046	0.192	−0.080	0.175
	mDLPFC9	0.067	0.034	0.155	−0.050	0.180	−0.113 ^#^	−0.076	0.207	−0.180	−0.029
	lFPC10	0.122	0.072	0.237	−0.036	0.218	0.008	−0.034	0.225	−0.141	0.185
	rFPC10	0.126	0.055	0.199	0.000	0.228	0.008	−0.020	0.179	−0.103	0.084
	mFPC10	−0.058	−0.074	0.250	−0.265	0.081	−0.131	−0.188	0.189	−0.213	0.051
Test HbR [µmol/L]	lDLPFC9	−0.020	−0.033	0.058	−0.066	0.007	0.015	0.009	0.051	−0.010	0.047
	rDLPFC9	−0.026	−0.029	0.043	−0.061	−0.002	0.009	0.009	0.052	−0.017	0.038
	lDLPFC46	−0.047	−0.028	0.061	−0.086	−0.008	0.011	0.004	0.074	−0.040	0.061
	rDLPFC46	−0.014	0.002	0.057	−0.066	0.035	0.006	0.010	0.061	−0.030	0.049
	mDLPFC9	−0.055	−0.055	0.065	−0.114	−0.001	0.000	−0.006	0.056	−0.031	0.030
	lFPC10	−0.048	−0.046	0.055	−0.084	0.003	−0.003	0.001	0.049	−0.044	0.043
	rFPC10	−0.023	−0.038	0.058	−0.067	0.020	0.011	0.002	0.064	−0.037	0.038
	mFPC10	−0.014	−0.015	0.075	−0.045	0.011	−0.004	0.003	0.062	−0.050	0.052
Retest HbO [µmol/L]	lDLPFC9	0.047	0.016	0.112	−0.026	0.128	−0.121	−0.107	0.237	−0.346	0.058
	rDLPFC9	0.099	0.035	0.221	−0.056	0.296	−0.114	−0.142	0.271	−0.216	0.060
	lDLPFC46	0.215	0.227	0.169	0.122	0.331	0.055	0.045	0.202	−0.066	0.244
	rDLPFC46	0.208	0.170	0.281	0.040	0.452	0.089	0.081	0.305	−0.036	0.236
	mDLPFC9	0.086	0.100	0.157	−0.046	0.215	−0.114	−0.119	0.258	−0.224	0.073
	lFPC10	0.166	0.118	0.195	0.004	0.324	−0.005	0.000	0.238	−0.198	0.073
	rFPC10	0.130	0.138	0.149	0.029	0.229	−0.031	−0.009	0.187	−0.107	0.071
	mFPC10	−0.050	−0.053	0.206	−0.173	0.090	−0.121	−0.098	0.179	−0.270	−0.016
Retest HbR [µmol/L]	lDLPFC9	0.005 ^#^	−0.028	0.070	−0.034	0.052	0.029	0.020	0.057	−0.014	0.058
	rDLPFC9	−0.028	−0.031	0.040	−0.067	0.008	0.010	0.008	0.076	−0.043	0.078
	lDLPFC46	−0.036	−0.050	0.070	−0.086	0.018	0.019	0.015	0.056	−0.005	0.044
	rDLPFC46	−0.081 ^#^	−0.073	0.122	−0.109	0.001	0.010	0.010	0.067	−0.038	0.063
	mDLPFC9	−0.054	−0.054	0.048	−0.108	−0.016	0.016	0.012	0.050	−0.013	0.051
	lFPC10	−0.056	−0.060	0.062	−0.091	−0.011	0.002	0.010	0.056	−0.043	0.052
	rFPC10	−0.049	−0.042	0.054	−0.088	−0.008	0.000	−0.008	0.069	−0.035	0.034
	mFPC10	−0.027	−0.021	0.072	−0.070	0.041	0.008	0.010	0.053	−0.020	0.044

MS: multiple sclerosis; HC: healthy control; MD: median; SD: standard deviation; HbO: oxyhemoglobin; HbR: deoxyhemoglobin; l/r/mDLPFC9: left/right/medial dorsolateral prefrontal cortex Brodmann area 9; l/rDLPFC46: left/right dorsolateral prefrontal cortex Brodmann area 46; l/r/mFPC10: left/right/medial frontopolar cortex Brodmann area 10; ^#^: not normally distributed.

**Table 3 brainsci-10-00643-t003:** Intraclass correlation coefficient of test and retest data of both groups (MS/HC).

Inter-Session Reliability		Mean							Slope Coefficient							Area Under the Curve						
			95% CI		F Test With True Value 0					95% CI		F Test With True Value 0					95% CI		F Test With True Value 0			
		ICC	Lower	Upper	Value	df1	df2	*p*	ICC	Lower	Upper	Value	df1	df2	*p*	ICC	Lower	Upper	Value	df1	df2	*p*
MS HbO (*n* = 16)	lDLPFC9	0.172^#^	−0.313	0.598	1.429	15	15	0.249	0.174°	−0.096	0.534	2.630	15	15	0.035	0.172^#^	−0.313	0.598	1.429	15	15	0.249
	rDLPFC9	0.026^#^	−0.490	0.510	1.050	15	15	0.463	0.136	−0.269	0.548	1.376	15	15	0.272	0.025^#^	−0.491	0.510	1.049	15	15	0.463
	lDLPFC46	−0.110^#^	−0.527	0.377	0.789	15	15	0.674	0.035	−0.370	0.477	1.082	15	15	0.440	−0.110^#^	−0.528	0.377	0.789	15	15	0.674
	rDLPFC46	−0.210	−0.688	0.331	0.674	15	15	0.773	0.101	−0.443	0.568	1.211	15	15	0.358	−0.208^#^	−0.687	0.333	0.677	15	15	0.771
	mDLPFC9	0.389	−0.134	0.736	2.208	15	15	0.068	0.234	−0.312	0.651	1.576	15	15	0.194	0.388	−0.134	0.736	2.206	15	15	0.068
	lFPC10	0.257	−0.272	0.661	1.667	15	15	0.167	**0.541**	0.066	0.813	3.227	15	15	0.015	0.256	−0.273	0.660	1.665	15	15	0.167
	rFPC10	0.313	−0.231	0.697	1.856	15	15	0.121	0.207	−0.326	0.631	1.500	15	15	0.221	**0.418**	−0.097	0.751	2.366	15	15	0.053
	mFPC10	0.157	−0.392	0.604	1.351	15	15	0.284	**0.579**	0.125	0.830	3.610	15	15	0.009	0.159	−0.390	0.605	1.354	15	15	0.282
MS HbR (*n* = 16)	lDLPFC9	0.360^#^	−0.113	0.711	2.190	15	15	0.070	0.097	−0.102	0.406	1.647	15	15	0.172	0.361	−0.112	0.712	2.193	15	15	0.070
	rDLPFC9	0.391	−0.138	0.739	2.206	15	15	0.068	0.093	−0.065	0.379	2.182	15	15	0.071	0.391	−0.138	0.739	2.207	15	15	0.068
	lDLPFC46	**0.561**	0.106	0.822	3.470	15	15	0.011	**0.398**	−0.116	0.740	2.267	15	15	0.062	**0.562**	0.107	0.822	3.477	15	15	0.011
	rDLPFC46	0.201^#^	−0.198	0.591	1.638	15	15	0.175	0.298	−0.155	0.669	1.951	15	15	0.104	0.202	−0.198	0.592	1.640	15	15	0.174
	mDLPFC9	**0.523**	0.036	0.805	3.060	15	15	0.019	0.158	−0.387	0.604	1.355	15	15	0.282	**0.523**	0.035	0.805	3.058	15	15	0.019
	lFPC10	**0.536**	0.065	0.810	3.212	15	15	0.015	**0.469**	−0.006	0.774	2.765	15	15	0.029	**0.538**	0.067	0.811	3.223	15	15	0.015
	rFPC10	**0.483**	0.042	0.777	3.191	15	15	0.016	**0.629**	0.232	0.850	4.722	15	15	0.002	**0.514**	0.073	0.794	3.581	15	15	0.009
	mFPC10	**0.464**	−0.029	0.774	2.670	15	15	0.033	**0.403**	−0.113	0.743	2.289	15	15	0.060	**0.463**	−0.030	0.773	2.665	15	15	0.033
HC HbO (*n* = 19)	lDLPFC9	**0.744**	0.457	0.892	6.841	18	18	0.000	**0.560**	0.162	0.803	3.535	18	18	0.005	**0.745**	0.457	0.892	6.845	18	18	0.000
	rDLPFC9	**0.529** **^#^**	0.099	0.790	3.132	18	18	0.010	**0.481**	0.070	0.758	3.301	18	18	0.008	**0.529** **^#^**	0.099	0.790	3.133	18	18	0.010
	lDLPFC46	**0.478**	0.048	0.760	2.813	18	18	0.017	0.208	−0.273	0.600	1.508	18	18	0.196	**0.478**	0.049	0.760	2.816	18	18	0.017
	rDLPFC46	**0.563**	0.151	0.807	3.461	18	18	0.006	0.306°	−0.124	0.652	1.944	18	18	0.084	**0.563**	0.151	0.807	3.466	18	18	0.006
	mDLPFC9	0.392^#^	−0.083	0.715	2.221	18	18	0.050	0.062	−0.385	0.489	1.133	18	18	0.397	0.392^#^	−0.083	0.715	2.220	18	18	0.050
	lFPC10	**0.612**	0.223	0.831	4.001	18	18	0.003	0.343°	−0.140	0.687	1.990	18	18	0.077	**0.612**	0.223	0.831	4.003	18	18	0.003
	rFPC10	**0.507**	0.087	0.775	3.040	18	18	0.012	**0.492**	0.054	0.769	2.861	18	18	0.016	0.368	−0.107	0.701	2.110	18	18	0.061
	mFPC10	**0.615**	0.228	0.833	4.044	18	18	0.002	**0.512**	0.074	0.781	2.989	18	18	0.013	**0.616**	0.228	0.833	4.046	18	18	0.002
HbR (*n* = 19)	lDLPFC9	**0.548**	0.153	0.796	3.488	18	18	0.006	**0.395**	−0.044	0.710	2.330	18	18	0.040	**0.548**	0.153	0.796	3.490	18	18	0.006
	rDLPFC9	**0.533**	0.122	0.789	3.258	18	18	0.008	**0.543**	0.134	0.795	3.334	18	18	0.007	**0.533**	0.122	0.789	3.261	18	18	0.008
	lDLPFC46	**0.552**	0.131	0.802	3.334	18	18	0.007	**0.479**	0.068	0.757	2.939	18	18	0.014	**0.552**	0.131	0.802	3.336	18	18	0.007
	rDLPFC46	**0.522**	0.090	0.786	3.076	18	18	0.011	0.331	−0.131	0.676	1.978	18	18	0.079	**0.522**	0.090	0.786	3.075	18	18	0.011
	mDLPFC9	**0.418**	−0.015	0.724	2.469	18	18	0.031	0.306	−0.179	0.664	1.841	18	18	0.102	**0.417**	−0.015	0.723	2.468	18	18	0.031
	lFPC10	**0.633**	0.259	0.841	4.309	18	18	0.002	0.123	−0.360	0.544	1.270	18	18	0.309	**0.633**	0.259	0.841	4.307	18	18	0.002
	rFPC10	**0.580**	0.186	0.814	3.704	18	18	0.004	**0.684**	0.341	0.865	5.137	18	18	0.001	**0.622** **^#^**	0.245	0.836	4.182	18	18	0.002
	mFPC10	0.392	−0.061	0.712	2.272	18	18	0.045	0.206	−0.210	0.581	1.572	18	18	0.173	0.392	−0.062	0.712	2.270	18	18	0.045

MS: multiple sclerosis; HC: healthy controls; ICC: intraclass correlation coefficient; CI: confidence interval; df: degree of freedom; *p*: *p*-value; HbO: oxyhemoglobin; HbR: deoxyhemoglobin; l/r/mDLPFC9: left/right/medial dorsolateral prefrontal cortex Brodmann area 9; l/rDLPFC46: left/right dorsolateral prefrontal cortex Brodmann area 46; l/r/mFPC10: left/right/medial frontopolar cortex Brodmann area 10; ^#^: not normally distributed; bold: fair–excellent intraclass correlation coefficient.

**Table 4 brainsci-10-00643-t004:** Bland and Altman limits of agreement, bias and confidence intervals of both groups (MS/HC).

**Bland and Altman** **(Test–Retest)**			**Mean**									**Area Under the Curve**								
						**Upper LoA**			**Lower LoA**						**Upper LoA**			**Lower LoA**		
		***n***	**Mean**	**SD**	**SE**	**LoA**	**Upper CI**	**Lower CI**	**LoA**	**Upper CI**	**Lower CI**	**Mean**	**SD**	**SE**	**LoA**	**Upper CI**	**Lower CI**	**LoA**	**Upper CI**	**Lower CI**
MS HbO [µmol/L]	rDLPFC9	16	−0.044	0.280	0.070	0.505	0.760	0.250	−0.592	−0.337	−0.847	−0.869	5.592	1.398	10.090	15.184	4.996	−11.829	−6.735	−16.923
	lDLPFC9	16	0.067	0.219	0.055	0.497	0.696	0.297	−0.362	−0.163	−0.562	1.341	4.378	1.094	9.921	13.909	5.933	−7.239	−3.251	−11.227
	rDLPFC46	16	0.003	0.339	0.085	0.667	0.976	0.358	−0.661	−0.353	−0.970	0.057	6.767	1.692	13.320	19.485	7.155	−13.207	−7.042	−19.372
	lDLPFC46	16	0.099	0.282	0.071	0.652	0.909	0.395	−0.454	−0.197	−0.710	1.986	5.639	1.410	13.039	18.176	7.901	−9.067	−3.930	−14.204
	mDLPFC9	16	−0.020	0.174	0.044	0.322	0.481	0.163	−0.361	−0.203	−0.520	−0.395	3.486	0.871	6.437	9.613	3.262	−7.227	−4.052	−10.403
	lFPC10	16	−0.044	0.266	0.066	0.477	0.719	0.235	−0.565	−0.323	−0.807	−0.879	5.309	1.327	9.526	14.363	4.690	−11.285	−6.449	−16.122
	rFPC10	16	−0.005	0.208	0.052	0.403	0.592	0.213	−0.412	−0.223	−0.601	−0.419	3.545	0.886	6.529	9.759	3.300	−7.368	−4.138	−10.597
	mFPC10	16	−0.008	0.299	0.075	0.578	0.851	0.306	−0.593	−0.321	−0.866	−0.152	5.967	1.492	11.543	16.978	6.107	−11.847	−6.411	−17.283
MS HbR [µmol/L]	rDLPFC9	16	0.002	0.046	0.012	0.092	0.134	0.050	−0.088	−0.046	−0.130	0.043	0.920	0.230	1.845	2.683	1.008	−1.759	−0.921	−2.597
	lDLPFC9	16	−0.025	0.072	0.018	0.116	0.181	0.050	−0.165	−0.100	−0.230	−0.497	1.431	0.358	2.308	3.612	1.004	−3.302	−1.998	−4.605
	rDLPFC46	16	0.067	0.117	0.029	0.297	0.404	0.190	−0.163	−0.056	−0.270	1.342	2.344	0.586	5.937	8.073	3.802	−3.253	−1.117	−5.388
	lDLPFC46	16	−0.010	0.062	0.016	0.112	0.169	0.055	−0.133	−0.076	−0.189	−0.206	1.246	0.312	2.237	3.373	1.102	−2.649	−1.513	−3.785
	mDLPFC9	16	−0.001	0.056	0.014	0.109	0.161	0.058	−0.112	−0.060	−0.163	−0.021	1.128	0.282	2.189	3.216	1.162	−2.231	−1.204	−3.258
	lFPC10	16	0.008	0.057	0.014	0.120	0.172	0.068	−0.104	−0.052	−0.156	0.156	1.142	0.285	2.394	3.434	1.354	−2.081	−1.041	−3.121
	rFPC10	16	0.026	0.054	0.014	0.133	0.182	0.083	−0.080	−0.031	−0.130	0.485	0.910	0.228	2.269	3.098	1.440	−1.299	−0.469	−2.128
	mFPC10	16	0.013	0.077	0.019	0.164	0.234	0.094	−0.138	−0.068	−0.208	0.258	1.543	0.386	3.283	4.688	1.877	−2.766	−1.360	−4.172
HC HbO [µmol/L]	rDLPFC9	19	0.008	0.244	0.056	0.486	0.689	0.282	−0.469	−0.265	−0.673	0.168	4.869	1.117	9.712	13.782	5.641	−9.375	−5.304	−13.445
	lDLPFC9	19	0.038	0.162	0.037	0.356	0.492	0.220	−0.280	−0.144	−0.416	0.763	3.243	0.744	7.119	9.829	4.408	−5.592	−2.882	−8.303
	rDLPFC46	19	−0.022	0.241	0.055	0.451	0.653	0.249	−0.495	−0.294	−0.697	−0.444	4.822	1.106	9.007	13.038	4.976	−9.895	−5.864	−13.926
	lDLPFC46	19	0.049	0.236	0.054	0.512	0.710	0.315	−0.414	−0.217	−0.612	0.979	4.723	1.084	10.237	14.186	6.288	−8.278	−4.330	−12.227
	mDLPFC9	19	0.000	0.260	0.060	0.511	0.728	0.293	−0.510	−0.292	−0.728	0.006	5.206	1.194	10.209	14.562	5.857	−10.198	−5.846	−14.550
	lFPC10	19	0.013	0.207	0.048	0.419	0.593	0.246	−0.393	−0.220	−0.567	0.262	4.144	0.951	8.384	11.849	4.920	−7.861	−4.397	−11.326
	rFPC10	19	0.038	0.182	0.042	0.396	0.548	0.243	−0.319	−0.167	−0.471	0.244	3.039	0.697	6.201	8.742	3.660	−5.713	−3.172	−8.254
	mFPC10	19	−0.010	0.164	0.038	0.311	0.448	0.174	−0.331	−0.194	−0.468	−0.196	3.272	0.751	6.218	8.954	3.483	−6.610	−3.874	−9.345
HC HbR [µmol/L]	rDLPFC9	19	−0.011	0.052	0.012	0.092	0.136	0.048	−0.114	−0.070	−0.157	−0.214	1.049	0.241	1.842	2.719	0.965	−2.270	−1.393	−3.147
	lDLPFC9	19	−0.014	0.051	0.012	0.085	0.128	0.043	−0.114	−0.071	−0.156	−0.285	1.015	0.233	1.704	2.552	0.855	−2.274	−1.426	−3.122
	rDLPFC46	19	−0.004	0.063	0.015	0.120	0.173	0.067	−0.128	−0.075	−0.181	−0.083	1.267	0.291	2.400	3.459	1.341	−2.565	−1.506	−3.624
	lDLPFC46	19	0.001	0.072	0.017	0.142	0.203	0.082	−0.140	−0.080	−0.201	0.019	1.443	0.331	2.846	4.052	1.640	−2.809	−1.603	−4.015
	mDLPFC9	19	−0.016	0.057	0.013	0.096	0.144	0.048	−0.128	−0.080	−0.175	−0.316	1.140	0.262	1.918	2.871	0.965	−2.551	−1.598	−3.504
	lFPC10	19	−0.005	0.045	0.010	0.084	0.122	0.046	−0.094	−0.056	−0.132	−0.103	0.907	0.208	1.674	2.432	0.916	−1.880	−1.122	−2.638
	rFPC10	19	0.011	0.061	0.014	0.131	0.182	0.080	−0.109	−0.058	−0.160	0.133	0.981	0.225	2.055	2.875	1.235	−1.790	−0.970	−2.610
	mFPC10	19	−0.012	0.064	0.015	0.113	0.166	0.059	−0.137	−0.084	−0.191	−0.247	1.275	0.292	2.251	3.317	1.186	−2.745	−1.680	−3.811
**Bland and Altman** **(Test–Retest)**		**Slope Coefficient**																		
						**Upper LoA**			**Lower LoA**											
		***n***	**Mean**	**SD**	**SE**	**LoA**	**upper CI**	**lower CI**	**LoA**	**upper CI**	**lower CI**									
MS HbO [µmol/L]	rDLPFC9	16	0.167	0.329	0.082	0.812	1.111	0.512	−0.478	−0.178	−0.777									
	lDLPF9	16	0.385	0.225	0.056	0.826	1.031	0.621	−0.055	0.149	−0.260									
	rDLPFC46	16	−0.014	0.417	0.104	0.803	1.183	0.423	−0.831	−0.452	−1.211									
	lDLPFC46	16	0.179	0.394	0.099	0.952	1.311	0.593	−0.594	−0.235	−0.953									
	mDLPFC9	16	0.028	0.404	0.101	0.818	1.186	0.451	−0.763	−0.396	−1.131									
	lFPC10	16	0.022	0.247	0.062	0.506	0.731	0.281	−0.461	−0.236	−0.686									
	rFPC10	16	−0.041	0.269	0.067	0.486	0.731	0.241	−0.568	−0.323	−0.813									
	mFPC10	16	−0.033	0.296	0.074	0.547	0.817	0.278	−0.613	−0.343	−0.883									
MS HbR [µmol/L]	rDLPFC9	16	−0.108	0.049	0.012	−0.012	0.033	−0.057	−0.205	−0.160	−0.249									
	lDLPFC9	16	−0.105	0.073	0.018	0.038	0.104	−0.029	−0.248	−0.182	−0.315									
	rDLPFC46	16	0.062	0.144	0.036	0.345	0.477	0.213	−0.221	−0.090	−0.353									
	lDLPFC46	16	−0.015	0.103	0.026	0.186	0.280	0.093	−0.217	−0.123	−0.311									
	mDLPFC9	16	0.009	0.115	0.029	0.233	0.338	0.129	−0.216	−0.112	−0.321									
	lFPC10	16	−0.018	0.072	0.018	0.123	0.189	0.058	−0.159	−0.093	−0.224									
	rFPC10	16	0.022	0.054	0.014	0.128	0.177	0.079	−0.085	−0.035	−0.134									
	mFPC10	16	0.013	0.098	0.025	0.205	0.294	0.116	−0.179	−0.090	−0.269									
HC HbO [µmol/L]	rDLPFC9	19	0.065	0.120	0.027	0.300	0.400	0.200	−0.169	−0.069	−0.270									
	lDLPF9	19	−0.029	0.129	0.030	0.224	0.331	0.116	−0.281	−0.173	−0.388									
	rDLPFC46	19	−0.107	0.305	0.070	0.490	0.745	0.236	−0.704	−0.449	−0.959									
	lDLPFC46	19	0.046	0.312	0.071	0.656	0.917	0.396	−0.565	−0.305	−0.826									
	mDLPFC9	19	−0.071	0.284	0.065	0.486	0.724	0.248	−0.629	−0.391	−0.867									
	lFPC10	19	0.010	0.245	0.056	0.490	0.694	0.285	−0.469	−0.265	−0.674									
	rFPC10	19	−0.018	0.154	0.035	0.284	0.413	0.155	−0.320	−0.191	−0.449									
	mFPC10	19	−0.001	0.198	0.045	0.387	0.552	0.221	−0.388	−0.223	−0.553									
HC HbR [µmol/L]	rDLPFC9	19	0.008	0.042	0.010	0.090	0.125	0.055	−0.074	−0.039	−0.109									
	lDLPF9	19	0.012	0.043	0.010	0.097	0.133	0.060	−0.073	−0.037	−0.110									
	rDLPFC46	19	−0.016	0.083	0.019	0.147	0.216	0.077	−0.180	−0.110	−0.249									
	lDLPFC46	19	−0.023	0.069	0.016	0.114	0.172	0.055	−0.159	−0.101	−0.217									
	mDLPFC9	19	0.006	0.074	0.017	0.151	0.213	0.089	−0.140	−0.078	−0.202									
	lFPC10	19	−0.009	0.070	0.016	0.128	0.186	0.070	−0.145	−0.087	−0.203									
	rFPC10	19	−0.005	0.045	0.010	0.084	0.121	0.046	−0.093	−0.055	−0.130									
	mFPC10	19	−0.021	0.054	0.012	0.085	0.131	0.040	−0.128	−0.082	−0.173									

MS: multiple sclerosis; HC: healthy control; CI: confidence interval; SD: standard deviation; SE: standard error; LoA: limits of agreement; HbO: oxyhemoglobin; HbR: deoxyhemoglobin; l/r/mDLPFC9: left/right/medial dorsolateral prefrontal cortex Brodmann area 9; l/rDLPFC46: left/right dorsolateral prefrontal cortex Brodmann area 46; l/r/mFPC10: left/right/medial frontopolar cortex Brodmann area 10.

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
