# Peer review of "Inter-Session Reliability of Functional Near-Infrared Spectroscopy at the Prefrontal Cortex While Walking in Multiple Sclerosis"

_brainsci, 2020, doi:10.3390/brainsci10090643_

Round 1

Reviewer 1 Report

This work could provide a fundamental reference for many other applications using fNIRS, which can be interesting to many researchers. This study is well designed. Very detailed data are presented. However, the following aspects could be improved to further improve the quality of the paper.

  1. ISR is never explicitly defined throughout the article. Since ISR is the major topic of this work, it is very essential to give a clear definition. Brief but clear description of ISR should also be inserted into abstract. 
  2. Figure 2. It is not clear which locations are sources(S) and which are detectors(D). Which S and D are linked to form S D pairs? This information should be provided for readers to understand which areas the measurements covered and to follow the methods. 

Reviewer 2 Report

In this study, inter-session reliability (ISR) of functional-near infrared spectroscopy (fNIRS) was investigated in MS patients and healthy controls. The topic is interesting, and the study was well designed and performed. However, this manuscript will benefit from modifying the presentation of results (Table 2-4) for better readability. Please highlight the key results rather than listing the entire results.  Please consider making graphs or plots based on the raw data for better presentation of the results.  

In addition, the calculation of ISR, which is a key component in this study, needs more explanation. It is mentioned that absolute or relative ISR was calculated, but not specific details were provided regarding this. How SC and A were calculated? What was the definition of the SC (what over what parameter)? Any example using a figure in method or result section regarding this will be greatly helpful. Any equations regarding this will be also very helpful.

Here are some minor comments:

Page 1 Line 24) What are the numbers, 9/46/10?

Page 4 Line 126) What is the name of first filter? Soft/hard thresholding or hanning filter? Needs more details or a reference. Please provide a specific name of the lowpass filter too.

Page 4 Line 130) Please provide a reference or details about Beer-Lambert-Law.

Page 3 Line 110, Figure 2) Please introduce the acronyms for the subarea. It will be great if it can be annotated in Figure 2 (right image).

What was the temporal resolution of fNIRS used in this study? I see the continuous line in Figure 3. If the temporal resolution was low, please indicate each discrete point marker in the plots in Figure 3.

Figure 3) What subarea was used to obtain this figure? If it was 1DLPFC46, please mention that in the figure caption.

(Table 2 and 3) There are degree sign for Mean HbO/HbR. What does it mean? Do these tables show the results with all patients at a specific time point or all patient over all time point during the 6 min walking? It seems the averaged value according to the method section. Please make it clear here.

No statistical comparison was made between two groups (MS vs. HC). It would be also interesting to show the p-values of ISR parameters between the groups.

Reviewer 3 Report

This is a very interesting manuscript which investigated the Inter-session reliability of fNIRS at the prefrontal cortex while walking in Multiple Sclerosis. The paper brings novelty to the field, and it is well designed. Although, the manuscript and mainly the analyses must be revised. Below, the authors can find my detailed comments.

Abstract

Line 21 - I suggest the use of "people with MS". It reads better. If the authors agree, please, change throughout the text.

Line 22 - Replace "20" with "Twenty".

Line 25 - Please, state what areas are the 9/46/10. This helps the lay reader.

Introduction

General comment - Please, break down the introduction. It feels like a very long paragraph. It makes easier for the reader.

Methods

Line 75 - Please, add group or individuals, to read "the HC group" or "HC individuals".

Lines 93 - 95 - Since people with MS present fatigue, I would expect they have a different perception of their effort and exhaustion throughout the test, even more given the fact they took 12:30 minutes to complete the tests. Since the authors did not measure between the attempts, I would include the lack of measure as a limitation.

Line 103 - Were the caps the same size? If no, how did the authors accounted for different head sizes? If yes, please, state the head sizes caps that the authors used. This may change the data collected.

Line 107 - "number 15". Is this a channel? Optode? Please, be specific.

Lines 107 - 108 - "in the middle between nasion to 107 inion and left preauricular to right preauricular point". Did the authors measured that? If yes, please state that in the text.

Lines 109-114 - Did the authors use fOLD (Zimeo-Morais et al., 2018)? If not, I recommend the authors to review this text including information about specificity and % of coverage by each channel.

Line 125 - "standard deviation". In the HOMER 2 script, does not SD mean source-detector separation? If yes, please correct.

Line 134 - "-10 -45 s". Does the author mean -10 to 45s?

Line 141 - "subareas". What subareas? Please, state them.

Results

Line 155 - Change "could be analysed" to "were analysed"

Table 2 - Please, do not abbreviate median as MD. It makes the readers' lives easier.

Line 193 - "ICC". Please, first define ICC in the text.

Line 211 - "LoA". Please, state what LoA means, before any abbreviation.

Discussion

Lines 244 - 249 - Have the authors tried to analyse just the first few walks? I believe that this results would be more reliable in few walks than in a tiring test for people with MS, such as the 6MWT. A different protocol, for example, 5 walking tests for 30 would perhaps have clinical significance, can be reproduced in the clinical settings, and perhaps will eliminate such variability. In order to verify that, the authors could analyse the difference between the average of first 3 walks and the average of the last 3 walks. I think this is really important to be done, instead of grouping all the walks throughout the test. 

Lines 251 - 254 - I would be more concerned with the mind wandering during the baseline data collection than the during the walking test. Did the authors control how the participants performed the baseline? Have in mind that simple walking for people with MS (even though they were not totally impaired), can be tiring, and can perhaps lead to increase in the use of attentional resources. 

Lines 269 - 272 - Did the authors measure using Cz as reference (from nose to inion (anteroposterior) and preauricular points (mediolateral)? Also, how did the authors deal with light interference? Finally, I recommend the authors to report the methodological reporting criteria as suggested by Pelicioni et al., 2019. This will give to the readers at least an idea how confounders were controlled in your study. 

Line 273 - Very good observation. It is well-known in the field of motor learning and motor control that familiarisation is crucial to eliminate the learning effect of trials. I suggest that the authors re-analyse the data: eliminating the first attempts of both groups and each date, as well as eliminate the last attempts (as I mentioned above, people with MS may have experienced fatigue). If the difference between the first and last trials are statistically significant, please, do not add the last trials in your calculation. You will not be manipulating your data, but you will be certifying that data with no contamination due to learning effect (first attempts) and fatigue (last attempts) are included in your analysis.

Lines 276 - 277 - I fully agree with the authors with the familiarization statement.

Lines 279 - 280 - I would not include the point (ii). We cannot guarantee that DT condition will eliminate this problem. Your data may become more variable because of that.

Lines 281 - 282 - A way of solving the point (iv) is to check Zimeo-Morais paper and use the technique fOLD to include only channels that were very sensitive (perhaps above 50%). Also, the use of a correct assessment and different caps solve this problem.

Line 282 - I agree with point (v). Always offer the first attempt (s) as practice, regardless if the test is new or not, to eliminate learning effect.

References:
Zimeo-Morais et al., 2018 - https://pubmed.ncbi.nlm.nih.gov/29463928/

Pelicioni et al., 2019 - https://pubmed.ncbi.nlm.nih.gov/31110922/

Round 2

Reviewer 3 Report

Dear authors,

Thanks for accepting my suggestions. The manuscript now reads better now that some details that were missing, were incorporated in the text. I still have a very important observation, please, ensure to address it. 

Create a table, or include in the text the sensitivity of each channel according the fOLD tool. Guilherme Zimeo Morais and colleagues made available a supplementary material where authors can check such information. Otherwise, the fOLD software allow the authors to do that. 

Once again, congratulations for the great work done.
